# WSI-GT: Pseudo-Label Guided Graph Transformer for Whole-Slide Histology

## Abstract

Whole-slide histology images (WSIs) can exceed 100k × 100k pixels, making direct pixel-level segmentation infeasible and requiring patch-level classification as a practical alternative. However, most approaches either treat patches independently, ignoring spatial and biological context, or rely on deep graph models that oversmooth, leading to loss of critical tissue details.

We present WSI-GT (Pseudo-Label Guided Graph Transformer), a simple yet effective architecture that addresses these challenges. WSI-GT combines a lightweight local graph convolution block for neighborhood feature aggregation with a pseudo-label guided attention mechanism that preserves intra-class variability and mitigates oversmoothing. To cope with sparse annotations, we introduce an area-weighted sampling strategy that balances class representation while maintaining tissue topology.

WSI-GT achieves a Macro F1 of 0.95 on PATH-DT-MSU WSS2v2, improving by up to 3 percentage points over tile-based CNNs and by about 2 points over strong graph baselines. It further generalizes well to the Placenta benchmark and standard graph node classification datasets, highlighting both clinical relevance and broader applicability. These results position WSI-GT as a practical and scalable solution for graph-based learning on extremely large images.

## 1 Introduction

Modern digital pathology has revolutionized histopathological analysis by enabling extremely high-resolution acquisition of whole-slide images (WSIs), which provide comprehensive visualization of tissue specimens at 40× magnification (Hu et al., 2023b; Rodriguez et al., 2022). Although deep learning methods with Convolutional Neural Networks (CNNs) models have demonstrated remarkable success in processing these structurally complex images (Wulczyn et al., 2020; Khvostikov et al., 2023; Sun et al., 2024), they often struggle to effectively model spatial relationships. Therefore, Graph neural networks (GNNs) are increasingly being used in histological image analysis due to their ability to model spatial relationships.

Current GNN applications in WSI analysis focus primarily on graph-level predictions (Pati et al., 2022; Wu et al., 2024) (e.g., single disease labels per slide) rather than fragment-level classification (Bazargani et al., 2024). Adnan et al. (2020) proposed a two-stage framework for whole-slide image (WSI) representation learning, which involves sampling fragments using a color-based method and employing GNNs to learn the relationships among the sampled fragments. This representation can be used for downstream classification tasks. Chan et al. (2023) formulated the WSI as a heterogeneous graph and introduced a new heterogeneous-graph edge attribute transformer (HEAT) to leverage edge and node heterogeneity during message aggregation. Zheng et al. (2022) utilized a Graph-Transformer that combines a graph-based representation of a WSI with features from a vision transformer to predict disease grade, achieving high accuracy across various datasets. SlideGraph+ (Lu et al., 2022) extracts representative features from image fragments and constructs a WSI-level graph representation to predict HER2 status in breast cancer.

Additionally, patch-based classification of WSIs is a crucial task for histological image diagnosis. Throughout this work, we refer to patches as small square image regions extracted from WSIs. This patch classification task can be transformed into segmentation of different structures and tissues through spatial aggregation, which is the key step for automatic analysis of whole slide images. An

example of a whole-slide image with partial polygonal annotations is shown in Fig. 1. CNNs have achieved great success in this task (Wulczyn et al., 2020; Khvostikov et al., 2023; Sun et al., 2024; Hou et al., 2016). However, these approaches handle patches individually, with little consideration of their topological or biological relationships. It is evident that patches from adjacent areas should often have the same labels, and the best way to model such positional information is by using a graph structure. Therefore, our objective is to build a Graph Neural Network (GNN) to improve the fragment classification accuracy of WSIs.

To the best of our knowledge, few studies have applied GNNs to this problem. The closest is Graph V-Net (Li et al., 2023), which introduces a hierarchical GNN with semi-supervised patch pre-training. However, its graph is built from large sliding windows, where most patches share the same label and many are unannotated, limiting the use of spatial context. Moreover, the hierarchical node-reduction/expansion design is heavily parameterized, making training and deployment difficult in practice.

In view of these limitations, we propose WSI-GT, a simple Graph Transformer with Pseudo-Label Attention for precise tissue fragment classification in WSIs. Our method uses lightweight graph convolution layers to aggregate neighborhood information and a novel sampling strategy that draws patches from multiple annotated regions rather than a single area. To counter over-globalizing and overfitting in deep GNNs (Wu et al., 2023), we employ two Graph Convolution Blocks and two Pseudo-Label Attention Blocks. Unlike works that rely on METIS clustering (Xing et al., 2024), we define clusters directly from pseudo-labels predicted by a pretrained patch encoder, and apply self-attention within clusters before fusing the resulting features with those from graph convolution.

We evaluate WSI-GT on the PATH-DT-MSU WSS2v2 dataset (Sun et al., 2024) for patch-level tissue classification, comparing against state-of-the-art patch- and graph-based networks. Our method achieves a Macro F1 of 0.95, improving by up to 3 points over patch baselines and by about 2 points over strong graph baselines. We further validate WSI-GT on the Placenta benchmark (Vanea et al., 2022) and on universal graph node classification datasets (PubMed (Yang et al., 2016), Actor, Deezer (Lim et al., 2021)), demonstrating its generality. Finally, we propose a sampling strategy that enables rough WSI segmentation.

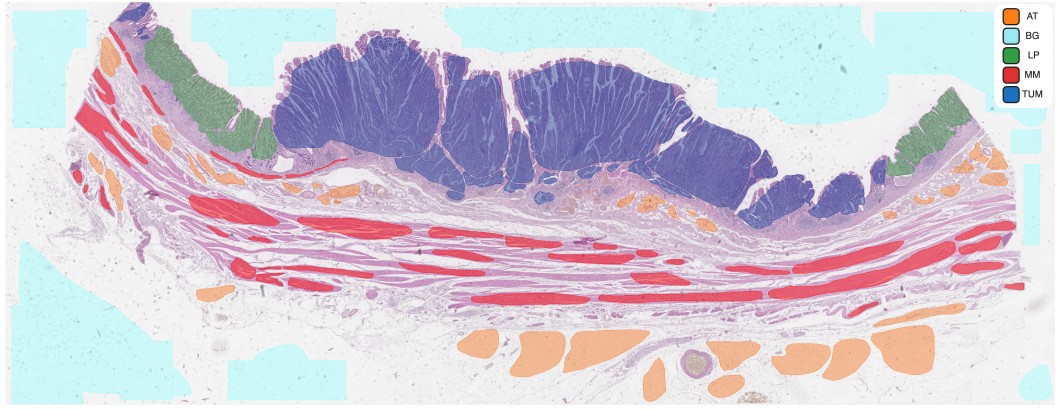

Figure 1: Example of a whole slide histological image from the PATH-DT-MSU WSS2v2 dataset with partial polygonal annotation of 5 tissue classes including background made by expert pathologists. The annotated areas are colored.

## 2 METHOD

### 2.1 SAMPLING GRAPH FROM ANNOTATED WSI

Due to GPU memory limitations, it is infeasible to load the full WSI graph into memory, so we sample subgraphs. Prior studies (Li et al., 2023; Zheng et al., 2022; Liang et al., 2023) usually construct graphs from fixed square regions using sliding windows, which has two drawbacks. First, not all patches in such regions are annotated; for example, Graph V-Net treated unannotated areas as an

additional class *normal*, which may introduce label noise since unannotated regions are not necessarily normal tissue. Second, graphs restricted to a single region often fail to capture meaningful spatial relationships: if tissue types change across boundaries, there is no exchange of information between regions, leading to misclassification.

To address these issues, we sample patches directly from annotated regions across the whole slide. To account for varying region sizes, we adopt the *Area-Weighted Random Sampling* method (Sun et al., 2024), where the probability of selecting region $i$ is proportional to its annotated area. This balances representation across classes while preserving tissue topology. Formally, the sampling probability is

$$
\begin{aligned}
p_i &= \hat{p}_i \Big/ \sum_{j=1}^N \hat{p}_j, \\
\hat{p}_i &= \begin{cases} \frac{1}{N} + \left( \frac{S_i^{-1}}{\sum_{j=1}^N S_j^{-1}} - \frac{1}{N} \right) \cdot c, & c \in [-1, 0], \\ \frac{1}{N} + \left( \frac{S_i}{\sum_{j=1}^N S_j} - \frac{1}{N} \right) \cdot c, & c \in [0, 1], \end{cases}
\end{aligned}
\tag{1}
$$

where $p_i$ is the probability of sampling region $i$, $N$ is the number of annotated regions for the class, $S_i$ is its area, and $c$ is a balancing coefficient. Setting $c = 0$ yields uniform sampling; $c = 1$ samples strictly in proportion to area. We use $c = 0.5$ as a compromise, reflecting region size while maintaining diversity. To handle class imbalance, each tissue class is chosen with equal probability when sampling regions, ensuring balanced exposure despite frequency differences across classes. The sampled patches are then used to construct the graph, avoiding both class and area imbalance and improving the use of topological context.

Each subgraph ($N$=64 nodes) is processed independently, with no edges across subgraphs. At inference, logits from patches appearing in multiple subgraphs are averaged, and slide-level metrics are derived from all patch predictions.

## 2.2 GRAPH CONSTRUCTION FROM PATCHES

The concept of utilizing patch-level features as nodes involves splitting a WSI into smaller patches and extracting features from each patch to serve as nodes. This approach is commonly used in graph-based WSI learning (Gao et al., 2021; Adnan et al., 2020; Li et al., 2018). This popularity can be attributed to two primary reasons: firstly, splitting a WSI into smaller regions aligns with traditional computer vision methods; secondly, the segmented patches can be directly used to extract features using a pre-trained model without requiring additional preprocessing. We initially divide the WSI into tissue patches of $224 \times 224$ pixels. Following previous works (Liang et al., 2023; Zheng et al., 2022), we employ the ResNet50 model (He et al., 2015) to encode each patch into a $d$-dimensional vector $\{h_i \in \mathbb{R}^d, i = 1, 2, \ldots, N\}$, where $N$ represents the number of nodes in a single graph. What sets our approach apart is that while previous works typically use ResNet50 pre-trained on ImageNet, we further fine-tune it on target dataset PATH-DT-MSU WSS2v2 to better capture the relevant features. Each feature vector is considered as a node in the WSI graph, and we compile these feature vectors into a feature node matrix $\{X \in \mathbb{R}^{N \times d}\}$.

While the node feature matrix captures the characteristics of individual patches, the interactions between patches are equally important. In histological images, cells and tissues exhibit inherent spatial and biological relationships, such as substance exchange and bioelectrical signaling. To model these interactions and quantify their biological relevance, we construct edges between nodes according to their Euclidean distance. The core assumption is that spatially closer nodes interact more strongly. To define the edges, we utilize the K-nearest neighbors (KNN) algorithm (Bai et al., 2022; Su et al., 2021; Zhou et al., 2019; Hu et al., 2023a):

$$
\mathcal{A}_{i,j} = \begin{cases} 1, & \text{if } j \in \text{KNN}_k(i) \text{ and } d(i, j) < d_{\max}, \\ 0, & \text{otherwise.} \end{cases}
\tag{2}
$$

Here, $k$ is the number of nearest neighbors in $\text{KNN}_k(i)$, and $d_{\max}$ is the maximum Euclidean distance threshold (Xiang & Wu, 2021; Martin-Gonzalez et al., 2021). The edge $\mathcal{A}_{ij} = 1$ indicates a connection between node $i$ and node $j$ and $\mathcal{A}_{ij} = 0$ when the Euclidean distance between nodes exceeds this threshold or is not close enough. This allows us to model the tumor micro-environment because biological interactions between cells and tissue regions are inherently distance-dependent.

To better demonstrate the difference in graph construction with previous patch-level classification work Graph V-Net, a comparison with our constructed graph is illustrated in Fig. 2.

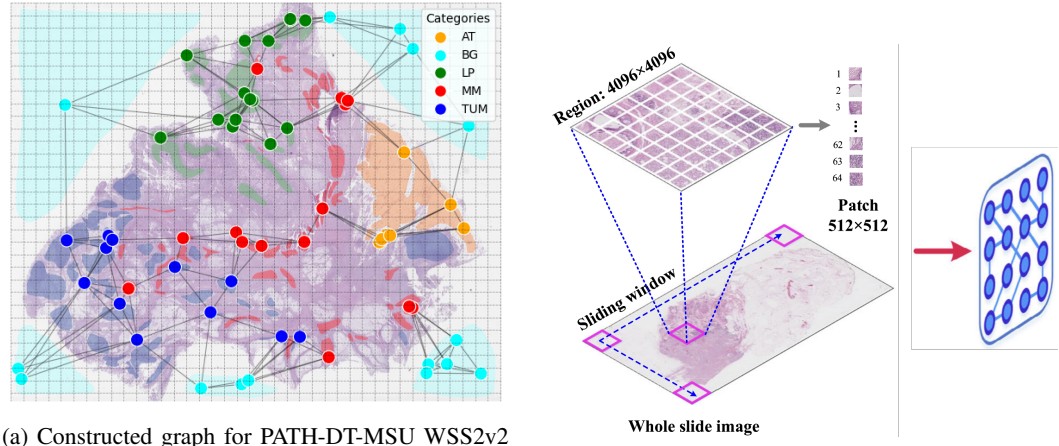

(a) Constructed graph for PATH-DT-MSU WSS2v2 using KNN ($k = 5$) and $N = 64$ with our method.

(b) Graph construction method from Graph V-Net.

Figure 2: Comparison of graph construction strategies.

## 2.3 WSI-GT Architecture

Let $X \in \mathbb{R}^{N \times d}$ be the matrix of patch features, $pl \in \{1, \dots, C\}^N$ the vector of pseudo-labels, and $P \in \mathbb{R}^{N \times d}$ the positional embeddings obtained from patch coordinates. We define $\tilde{X} = X + P$. Given adjacency $\mathcal{N}$, the outputs of WSI-GT for a sampled subgraph are

$$H_{\text{PLA}} = \text{PLA}(X, pl), \quad H_{\text{GC}} = \text{GC}(\tilde{X}, \mathcal{N}),$$
$$Z = (1 - \lambda) H_{\text{PLA}} + \lambda H_{\text{GC}}, \quad Y = \text{MLP}(Z), \tag{3}$$

where $\lambda \in [0, 1]$ is a mixing hyper-parameter (set to $0.8$ in our experiments, following (Wu et al., 2023)). $H_{\text{PLA}}, H_{\text{GC}} \in \mathbb{R}^{N \times d}$ are the updated node representations produced by the Pseudo-Label Attention and Graph Convolution blocks, respectively. For node $u$, $Z_u$ and $Y_u$ denote the corresponding rows of $Z$ and $Y$, giving its mixed embedding and final prediction. The overall training pipeline is illustrated in Figure 3. In our experiments we use $d = 2048$.

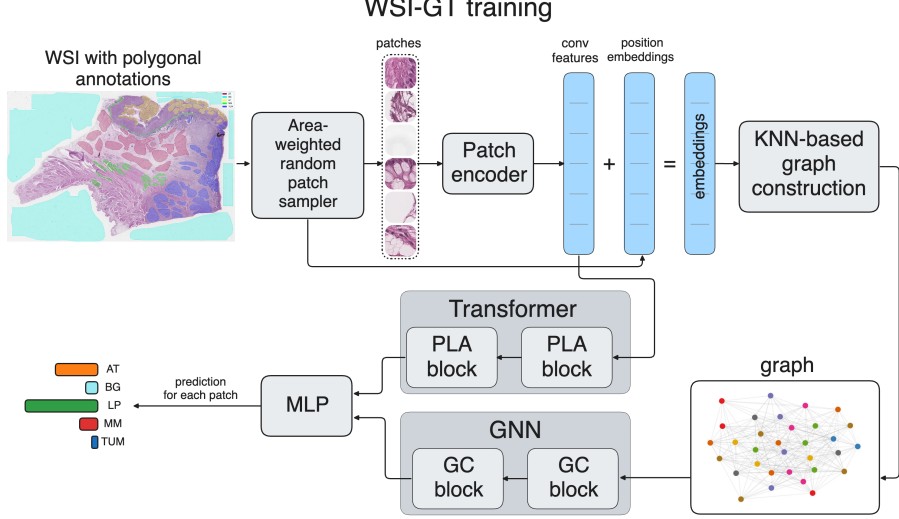

Figure 3: Overall training pipeline of WSI-GT.

### 2.3.1 GRAPH CONVOLUTION BLOCK

We employ a lightweight graph convolution block as the local module to aggregate information from neighboring nodes. For a node $u$, the output is

$$GC(\tilde{X}_u, \mathcal{N}_u) \;=\; \sigma\left(\sum_{v \in \mathcal{N}_u} \alpha_{uv} W \tilde{X}_v\right) + \tilde{X}_u, \tag{4}$$

where $\sigma(\cdot)$ denotes the ReLU activation. Here $\tilde{X}_u = X_u + p_u$ combines the patch feature $X_u$ (from the encoder) with its positional embedding $p_u$, generated from patch coordinates via a single linear layer; this leverages spatial priors (e.g., background patches often appear near WSI borders). The addition of $\tilde{X}_u$ implements a residual connection to stabilize training.

The attention weights $\alpha_{uv}$ follow the standard GAT formulation:

$$e_{uv} = \mathrm{LeakyReLU}\Big(a^\top \big[W\tilde{X}_u \,\|\, W\tilde{X}_v\big]\Big), \qquad \alpha_{uv} = \frac{\exp(e_{uv})}{\sum_{k \in \mathcal{N}_u} \exp(e_{uk})}. \tag{5}$$

Here $\|$ denotes concatenation, $W \in \mathbb{R}^{d \times d}$ is a learnable projection, and $a \in \mathbb{R}^{2d}$ maps the concatenated vector to a scalar logit $e_{uv}$. Note that $\alpha_{uv}$ (edge attention) is unrelated to the mixing parameter $\lambda$.

### 2.3.2 PSEUDO-LABEL ATTENTION BLOCK

To mitigate over-globalizing, we restrict self-attention to nodes sharing the same pseudo-label. Given node features $X \in \mathbb{R}^{N \times d}$, we use multi-head masked self-attention with $H = 8$ heads. For head $h \in \{1, \dots, H\}$ we compute

$$Q^{(h)} = XW_Q^{(h)}, \;\; K^{(h)} = XW_K^{(h)}, \;\; V^{(h)} = XW_V^{(h)},$$
$$S^{(h)} = \frac{Q^{(h)}(K^{(h)})^\top}{\sqrt{d_k}} + M, \qquad P^{(h)} = \mathrm{Softmax}_{\mathrm{row}}(S^{(h)}), \tag{6}$$
$$H^{(h)} = P^{(h)}V^{(h)},$$

where $W_Q^{(h)}, W_K^{(h)}, W_V^{(h)} \in \mathbb{R}^{d \times d_k}$. The mask $M \in \mathbb{R}^{N \times N}$ enforces intra-class attention:

$$M_{uv} = \begin{cases} 0, & pl_u = pl_v, \\ -\infty, & \text{otherwise.} \end{cases}$$

Outputs from all heads are concatenated and projected back to the model dimension,

$$H_{\mathrm{attn}} = \big[\, H^{(1)} \,\|\, \cdots \,\|\, H^{(H)} \,\big] W_O, \qquad W_O \in \mathbb{R}^{(H\,d_k) \times d}, \tag{7}$$

and a position-wise feed-forward layer yields the final output

$$H_{\mathrm{PLA}} = \mathrm{FFN}(H_{\mathrm{attn}}). \tag{8}$$

In our experiments we use $H{=}8$, $d{=}2048$, and $d_k{=}2048$.

## 2.4 INFERENCE FOR UNANNOTATED WSIs

Unlike dense graph sampling methods that use all regions, our random sampling approach based on annotated areas cannot be directly applied during whole-slide image (WSI) inference, as most regions are unannotated. To address this, we designed an Adaptive Coverage Sampler that efficiently and unbiasedly constructs graphs from unannotated WSIs by iteratively selecting batches of patches based on pixel-level coverage statistics. Each selected batch of patches forms a graph following the method described in Section 2.2, which is then passed to the network for prediction. After the network processes each batch, the algorithm checks whether every pixel in the WSI has been covered at least $\eta$ times – meaning each pixel has received predictions from $\eta$ overlapping patches. Once this $\eta$-coverage criterion is satisfied (in this work we use $\eta = 2$), the accumulated probability vectors at each pixel are aggregated to produce the final segmentation.

In summary, the Adaptive Coverage Sampler offers a robust, efficient, and unbiased sampling strategy suitable for WSIs with both annotated and unannotated areas. This ensures accurate predictions across the entire slide, leading to more reliable and comprehensive diagnostic outcomes.

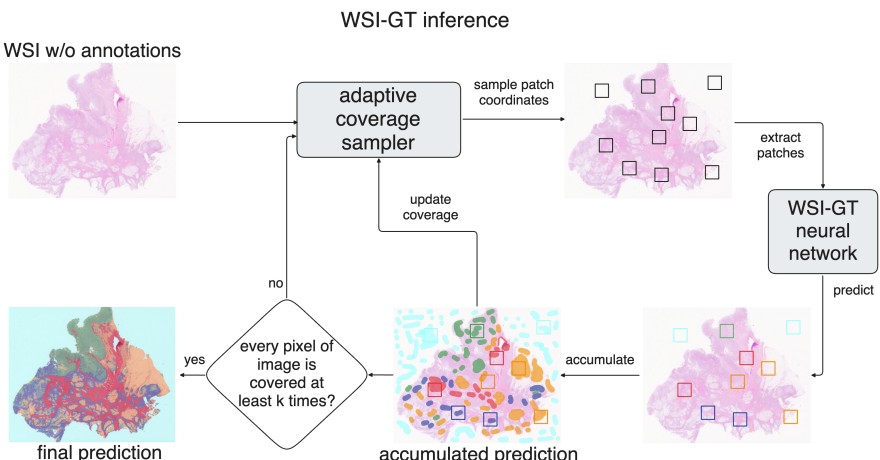

Figure 4: Preview of inference pipeline of WSI-GT.

## 3 DATASET AND EXPERIMENTS SETTING

### 3.1 DATASETS AND ANNOTATIONS

For patch-level classification, we use the PATH-DT-MSU WSS2v2 dataset (Sun et al., 2024), which provides polygonal annotations of gastric tissue types. It consists of 10 gastric cancer WSIs scanned at 40× magnification (average resolution ∼110k×90k pixels). Each slide is partially annotated by pathologists, with regions covering five tissue classes: adenocarcinoma (TUM), lamina propria (LP), muscularis mucosae (MM), submucosa/muscle/subserosa (AT), and background (BG). Figure 1 shows an example WSI with annotations.

We do not use the datasets from Graph V-Net (Li et al., 2023) for three reasons: (i) they combine the public BACH (Aresta et al., 2019) dataset with an unpublished set, limiting reproducibility; (ii) the modified BACH annotations are coarse and the test set lacks the "benign" class; (iii) BACH slides typically contain only one or two categories, making graph construction less meaningful. In contrast, PATH-DT-MSU WSS2v2 slides contain 4–5 categories with fine polygonal annotations, enabling richer evaluation. All competing methods were trained on the same annotated subset and identical train/validation/test splits as WSI-GT to ensure fairness.

To further validate generalizability, we evaluate WSI-GT on the Placenta cell-graph benchmark (Vanea et al., 2022) and on standard node classification datasets PubMed (Yang et al., 2016), Actor, and Deezer (Lim et al., 2021).

This diverse evaluation assesses both patch- and cell-level performance as well as applicability to generic graph benchmarks (results in Section 4).

### 3.2 IMPLEMENTATION DETAILS

We implemented our models in Python 3 with PyTorch and trained on a single NVIDIA A6000 (48GB). For patch-level classification, we used the PATH-DT-MSU WSS2v2 dataset with the sampling strategy from Section 2.1, applied consistently to training, validation, and test sets. Whole-slide segmentation followed the protocol in Section 2.4.

All experiments shared the same data augmentation, learning-rate schedules, and subgraph construction; only the architecture varied, ensuring that differences in performance reflect architectural design. Ablation studies, presented in the appendix, quantify the contribution of each component.

For cell-level classification[1] and general node classification benchmarks[2], we adopted the original evaluation pipelines, integrating our method to ensure comparability with prior results.

---

[1]https://github.com/Nellaker-group/placenta
[2]https://github.com/qitianwu/SGFormer

## 3.3 Comparison Methods and Evaluation Metrics

In the context of patch-level classification tasks, we used the F1 score and Macro F1 score to compare the proposed WSI-GT model with several widely used CNN architectures, including ResNet (He et al., 2015), DenseNet (Huang et al., 2017), EfficientNet (Tan & Le, 2021), and MobileNet (Howard et al., 2017). Additionally, the performance of WSI-GT was benchmarked against the previous best patch-based model SR+CLS (Sun et al., 2024). Furthermore, comparisons were made with several state-of-the-art graph neural networks, such as SAGEConv (Hamilton et al., 2017), Graph V-Net (Li et al., 2023), EdgeConv2d (Wang et al., 2019), MRConv2d (Li et al., 2019), and SGFormer (Wu et al., 2023). For cell-level classification tasks, the comparison was made with the original model results demonstrated in Placenta (Vanea et al., 2022). In the case of universal graph model classification, the results were directly taken from SGFormer (Wu et al., 2023).

## 4 Experiments and Results

### 4.1 Comparison on PATH-DT-MSU WSS2v2 Dataset

To validate the effectiveness of our proposed method, we conducted comprehensive comparisons with both patch-based and graph-based models. All models were evaluated under identical conditions: patch-based models were fine-tuned on the PATH-DT-MSU WSS2v2 dataset, while graph-based models employed the same graph sampling strategy. We selected optimal checkpoints based on validation performance and reported final results on the test set.

Table 1: Performance comparison of tissue type classification on the PATH-DT-MSU WSS2v2 dataset across different architectures.

| Method | Architecture | F1-score by class | | | | | Macro F1 |
| --- | --- | --- | --- | --- | --- | --- | --- |
| | | AT | BG | LP | MM | TUM | |
| ResNet50 (He et al., 2015) | Patch-based | 0.90 | 1.00 | 0.89 | 0.94 | 0.84 | 0.91 |
| DenseNet121 (Huang et al., 2017) | Patch-based | 0.93 | 1.00 | 0.82 | 0.94 | 0.83 | 0.90 |
| EfficientNet (Tan & Le, 2021) | Patch-based | 0.91 | 1.00 | 0.82 | 0.94 | 0.81 | 0.90 |
| MobileNet (Howard et al., 2017) | Patch-based | 0.88 | 1.00 | 0.80 | 0.91 | 0.79 | 0.88 |
| SR+CLS (Sun et al., 2024) | Patch-based | 0.91 | 1.00 | 0.88 | 0.94 | 0.85 | 0.92 |
| Graph V-Net (Li et al., 2023) | Graph-based | 0.95 | 1.00 | 0.90 | 0.91 | 0.89 | 0.93 |
| GraphSAGE (Hamilton et al., 2017) | Graph-based | 0.93 | 1.00 | 0.91 | 0.95 | 0.87 | 0.93 |
| EdgeConv2d (Wang et al., 2019) | Graph-based | 0.94 | 1.00 | 0.88 | 0.96 | 0.88 | 0.93 |
| MRConv2d (Li et al., 2019) | Graph-based | 0.94 | 1.00 | 0.87 | 0.96 | 0.86 | 0.92 |
| SGFormer (Wu et al., 2023) | Graph-based | 0.92 | 1.00 | 0.91 | 0.96 | 0.86 | 0.93 |
| **WSI-GT (Ours)** | Graph-based | **0.96** | **1.00** | **0.92** | **0.96** | **0.90** | **0.95** |

As shown in Table 1, WSI-GT achieves a Macro F1 of 0.95, representing a +3.0-point improvement over the best patch baseline (SR+CLS, 0.92) and +2.0 points over the best graph baseline (SGFormer, 0.93). This performance demonstrates the effectiveness of WSI-GT's architecture in combining local patch features with global spatial context. Figure 5 illustrates how WSI-GT leverages neighborhood information for correct classification, whereas patch-based models, which treat patches independently, often yield misclassifications. These errors are subsequently amplified in the overall coarse segmentation Figure 6.

### 4.2 Clinical Application Evaluation

The ultimate goal of patch-level classification is to enable semantic segmentation of entire whole-slide images (WSIs) for clinical analysis. Figure 6 compares our model's segmentation results with ResNet50 predictions, highlighting WSI-GT's superior performance.

To address the lack of ground truth in unannotated regions of WSIs, expert histopathological evaluation was performed by the medical co-authors. This evaluation confirmed several key findings:

- Tumor regions (TUM, blue) were accurately identified across tissue layers, capturing diffuse infiltration patterns,

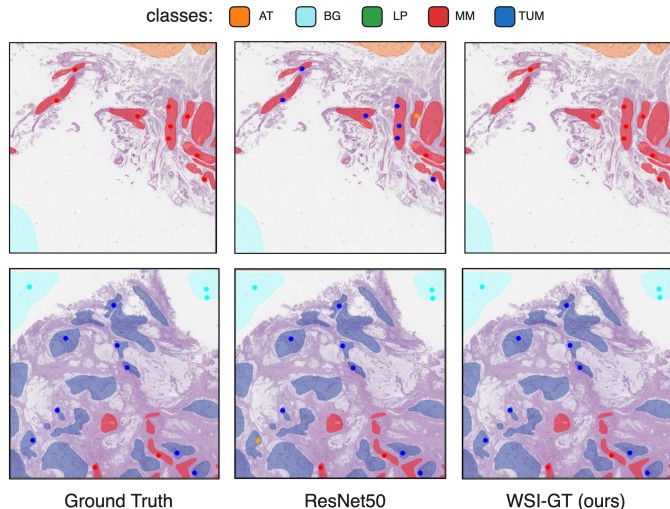

Figure 5: Qualitative results demonstrating WSI-GT's effective utilization of spatial context for accurate patch classification compared to CNN method (ResNet50). The points are the predictions made by networks and the colored regions are annotated regions with labels.

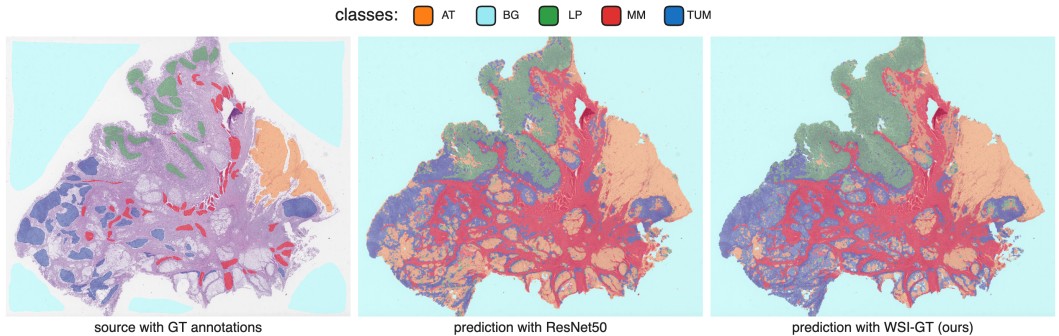

Figure 6: Comparative visualization of semantic segmentation results on PATH-DT-MSU WSS2v2 test_01 image. The ground-truth polygonal annotations were made by expert pathologists.

- lamina propria (LP, green) and muscularis mucosae (MM, red) showed precise segmentation despite tumor-induced morphological changes,
- Minor TUM-LP overlaps were clinically justified by shared mucus characteristics,
- Adipose tissue (AT, orange) was consistently correctly classified.

This clinical validation confirms that WSI-GT produces histologically plausible results, demonstrating its potential for real-world diagnostic applications.

### 4.3 COMPARISON ON CELL-LEVEL AND UNIVERSAL GRAPH NODE CLASSIFICATION BENCHMARKS

To validate the generalizability of WSI-GT beyond histological patch classification, we evaluated its performance on both specialized cell-level graphs and standard graph benchmarks. As shown in Table 2, WSI-GT achieves state-of-the-art performance on the Placenta cell-graph dataset with 64.98% accuracy, outperforming competitive baselines. WSI-GT maintains superior performance in the primary accuracy metric and demonstrates balanced results across all evaluation criteria.

The universal graph benchmarks in Table 3 reveal WSI-GT's consistent competitiveness across diverse domains. On the PubMed citation benchmark, our model achieves 80.6% accuracy, surpassing SGFormer by 0.3%. For the challenging Actor social network dataset, WSI-GT's 38.2% accuracy

Table 2: Cell-graph classification performance on the Placenta benchmark (Vanea et al., 2022).

| Model | Accuracy (%) | ROC AUC |
|---|---|---|
| **WSI-GT (Ours)** | **64.98** | 0.888 |
| GraphSAGE-mean (Hamilton et al., 2017) | 64.88 | 0.883 |
| SIGN (Rossi et al., 2020) | 64.77 | 0.886 |
| ClusterGCN (Chiang et al., 2019) | 64.24 | 0.882 |
| GraphSAINT-rw (Zeng et al., 2019) | 63.94 | **0.895** |
| ShaDow (Zeng et al., 2021) | 63.04 | 0.863 |
| ClusterGAT (Veličković et al., 2017) | 58.07 | 0.851 |
| ClusterGATv2 (Brody et al., 2021) | 57.07 | 0.854 |
| MLP Baseline | 47.98 | 0.750 |

Table 3: Node classification accuracy (%) on standard benchmarks. "# Nodes" and "# Edges" refer to the canonical dataset graphs.

| Model | PubMed | Actor | Deezer |
|---|---|---|---|
| **WSI-GT (Ours)** | **80.6** | **38.2** | 66.9 |
| CobFormer (Xing et al., 2024) | 80.5 | 37.4 | 66.9 |
| SGFormer (Wu et al., 2023) | 80.3 | 37.9 | **67.1** |
| APPNP (Klicpera, 2019) | 80.1 | 31.3 | 66.1 |
| SIGN (Rossi et al., 2020) | 79.5 | 36.5 | 66.3 |
| ClusterGAT (Veličković et al., 2017) | 79.0 | 29.8 | 61.7 |
| GCN (Kipf & Welling, 2017) | 78.8 | 30.1 | 62.7 |
| # Nodes | 19,717 | 7,600 | 28,281 |
| # Edges | 44,324 | 29,926 | 92,752 |

represents a 0.8% improvement over SIGN, the previous best-performing method. The Deezer results show near-parity between WSI-GT and CobFormer (Xing et al., 2024). These results collectively demonstrate that WSI-GT's architecture, particularly its pseudo-label attention mechanism, maintains strong performance across fundamentally different graph types - from biological cell-graphs to social networks and recommendation systems.

## 5 CONCLUSION

In this study, we proposed WSI-GT, a novel graph transformer architecture with pseudo-label attention and a histological graph sampling strategy to enhance fragment classification in histological whole-slide images (WSIs). Our method effectively integrates local structural information and global intra-class dependencies, addressing key challenges such as over-smoothing in deep graph neural networks and limited annotated data.

Extensive experiments on multiple datasets, including PATH-DT-MSU WSS2v2, Placenta, and standard graph node classification benchmarks, demonstrated the superior performance of WSI-GT. Expert evaluation by histopathologists within gastric WSIs confirmed that WSI-GT produces histologically plausible results, accurately identifying tumor regions across tissue layers and precisely segmenting challenging structures such as lamina propria and muscularis mucosae, despite morphology alterations induced by the disease. Additionally, WSI-GT showed strong performance on diverse benchmarks, further validating its effectiveness and generalizability.

In summary, WSI-GT offers a robust and effective solution for fragment classification in histological WSIs with potential applications in digital pathology. Moreover, due to its design, the method offers a promising solution not only for histology but also for a wide range of high-resolution image analysis tasks where exhaustive full annotation is prohibitively labor-intensive.

## REPRODUCIBILITY STATEMENT

All datasets used in this work are publicly available: PATH-DT-MSU WSS2v2 (Sun et al., 2024), Placenta (Vanea et al., 2022), PubMed (Yang et al., 2016), Actor and Deezer (Lim et al., 2021). For PATH-DT-MSU WSS2v2, we strictly follow the official dataset split and annotations provided by the dataset authors. Our implementation is based on Python 3 and PyTorch, and all experiments were conducted on a single NVIDIA A6000 GPU (48GB). Hyperparameters, training schedules, and data augmentation strategies are detailed in Appendix A.4, while ablation studies of graph sampling and architectural depth are presented in Appendix A.2. To facilitate reproducibility, we provide an anonymous repository containing code and training scripts at `https://anonymous.4open.science/r/wsi-gt-E7FB`. The full repository will be made public upon acceptance.

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

# A APPENDIX

This appendix provides additional details and experimental results to complement the main manuscript on graph neural networks (GNNs) for the classification of histological image fragments. All optimal parameters reported were identified through extensive experimentation on the PATH-DT-MSU WSS2v2 dataset of whole slide images (WSIs) with partial polygonal annotations. We present an in-depth analysis of two key components: (1) graph sampling strategies, comparing our Graph Sampling Strategy with Single-Region Sampling (Dense), demonstrating that Area-Weighted Random Sampling achieves superior performance (Macro F1: 0.95 vs. 0.91) by better capturing tissue heterogeneity; and (2) architecture depth, evaluating the impact of varying numbers of Graph Convolution (GC) and Pseudo-Label Attention (PLA) blocks. Our experiments reveal that a balanced configuration of 2 GC and 2 PLA blocks yields optimal performance (Macro F1: 0.95) on the used WSI dataset, while deeper architectures suffer from diminishing returns. These findings highlight the importance of sampling strategies and model depth in GNN-based histopathology analysis, offering practical insights for improving classification accuracy while mitigating over-globalization and computational inefficiencies. We also demonstrate our training hyperparameters and additional semantic segmentation visualizations for all WSIs in the test set.

## A.1 ADDITIONAL RELATED WORK

This section extends the literature review presented in our main paper, providing a more comprehensive overview of research applying GNNs to histological classification tasks. Given the diverse nature of these tasks, we categorize the related works into two main types: graph-level and patch-level classification, expanding upon the foundational studies discussed in the main text.

### A.1.1 GRAPH NEURAL NETWORKS FOR HISTOLOGICAL IMAGE ANALYSIS IN GRAPH-LEVEL CLASSIFICATION

In histological image analysis, graph-level classification using GNNs has attracted considerable attention because of its capability to capture the global structure and topological information within WSIs (Brussee et al., 2025). This approach is particularly advantageous for tasks such as cancer grading, survival prediction, and region-of-interest (ROI) classification, where the overall tissue structure and spatial relationships between different regions are crucial for accurate diagnosis. For instance, Zhou et al. (2019) proposed the CGC-Net, which uses a cell graph convolutional network to grade colorectal cancer histology images. This method leverages the spatial relationships between cells to provide a more accurate grading compared to traditional CNN-based approaches. Similarly, Wang et al. Wang et al. (2021) introduced a hierarchical graph pathomic network that integrates appearance, microenvironment, and topology for progression-free survival prediction. In addition, hierarchical GNNs have shown promise in graph-level classification tasks. For example, Pati et al. (2020) introduced HACT-Net, a hierarchical cell-to-tissue graph neural network that models the relationships between cells and tissues to improve classification accuracy. This hierarchical structure allows the model to capture multi-scale information, which is essential for tasks that require both fine-grained and global context. Overall, GNNs have proven to be effective in graph-level classification tasks in histological image analysis by leveraging the topological structure of WSIs. These models can capture complex spatial dependencies and provide more accurate predictions compared to traditional deep learning methods.

While graph-level classification using GNNs has demonstrated promising results, patch-level classification in histopathology introduces distinct challenges and opportunities. Key issues include processing whole-slide images (WSIs) at extremely high resolutions, mitigating patch-level misclassification, and preserving fine-grained spatial details. Despite these needs, graph-based methods for accurate patch-type classification in WSIs remain underexplored, representing a critical gap in computational pathology research.

### A.1.2 GRAPH NEURAL NETWORKS FOR HISTOLOGICAL IMAGE ANALYSIS IN PATCH-LEVEL CLASSIFICATION

Patch-level classification in histopathology involves predicting labels for individual patches within a WSI. This is crucial for tasks such as tissue semantic segmentation. Due to the large resolution of

Table 4: Effect of different graph construction strategies on macro F1, recall and accuracy, where "KNN" means using our strategy with different target and k, and "Dense" means constructing graph from large square moving windows.

| Graph sampling strategy | Macro F1 | Recall | Accuracy |
|---|---|---|---|
| KNN by features ($k = 3$) | 0.93 | 0.93 | 0.93 |
| KNN by features ($k = 5$) | 0.93 | 0.94 | 0.93 |
| KNN by distance ($k = 3$) | 0.94 | 0.94 | 0.93 |
| KNN by distance ($k = 5$) | **0.95** | **0.95** | **0.95** |
| KNN by distance ($k = 7$) | 0.94 | 0.95 | 0.95 |
| Dense (Li et al., 2023) | 0.92 | 0.91 | 0.92 |

WSIs, it is challenging to process the entire image directly and get a pixel-level mask prediction using CNNs. As a result, most existing methods apply CNNs to individual image patches (Khvostikov et al., 2021; Shen et al., 2022). Although Graph Neural Networks (GNNs) have been extensively employed for graph-level predictions, their utilization in patch-level classification is still relatively limited. SegGini (Anklin et al., 2021) is a notable example that constructs a tissue-graph representation from graph nodes and performs weakly-supervised segmentation via node classification using inexact image-level labels. Another relevant work is Graph V-Net (Li et al., 2023), which applies a hierarchical GNN with semi-supervised pre-training for histological image breast cancer classification.

However, these methods often use large sliding windows to build graphs, which may not fully utilize the spatial and contextual information within the patches. Moreover, deep and hierarchical GNNs can lead to additional computational costs and over-globalizing problems (Xing et al., 2024).

Despite these advances, key challenges persist: (1) limited exploitation of spatial-contextual dependencies between patches, (2) high computational overhead from complex GNN architectures, and (3) over-globalization degrading local discriminative power. Our proposed WSI-GT addresses these gaps by introducing a spatially aware graph construction strategy and a lightweight GNN design, effectively balancing local patch-level precision with global context. This approach not only reduces computational costs, but also mitigates over-globalization, offering a scalable solution for digital pathology tasks where patch-level accuracy is paramount.

## A.2 ABLATION STUDIES

### A.2.1 ANALYSIS OF GRAPH SAMPLING STRATEGIES

We compare two graph sampling strategies: (1) our proposed *Area-Weighted Random Sampling* method, which samples patches from annotated regions depending on their area, and (2) a *Single-Region Dense Sampling* baseline, which constructs graphs from sliding square windows and has been widely used in previous works (Li et al., 2023; Zheng et al., 2022; Liang et al., 2023).

For our method, we further investigate two key parameters in graph construction: the type of K-nearest neighbors (KNN) used to define edges (based on either feature similarity or spatial distance), and the number of neighbors $k$. The corresponding results are summarized in Table 4.

We observe that building graphs based on spatial distance (KNN by distance) yields better results compared to using feature similarity (KNN by features) or a fully connected approach. Specifically, KNN by distance with $k = 5$ achieves the best performance across all metrics. This suggests that incorporating spatial information during graph construction allows the model to better capture the contextual relationships between neighboring patches inside the whole slide image.

In contrast, using feature similarity for KNN results in slightly lower performance. This might be because relying solely on feature similarity might not accurately represent the spatial dependencies between patches, especially in complex tissue structures.

Finally, the Dense graph sampling method, often employed in existing works, exhibits the lowest performance. This highlights the potential drawbacks of connecting all patches indiscriminately, as it can introduce noise and irrelevant connections that hinder the GNN's ability to learn meaningful relationships.

Table 5: Effect of different numbers of Graph Convolution (GC) and Pseudo-Label Attention (PLA) blocks on macro F1, recall, and accuracy for the PATH-DT-MSU WSS2v2 dataset.

| GC blocks | PLA blocks | Macro F1 | Recall | Accuracy |
|---|---|---|---|---|
| 0 | 0 | 0.91 | 0.89 | 0.90 |
| 2 | 0 | 0.92 | 0.92 | 0.92 |
| 2 | 1 | 0.95 | 0.94 | 0.94 |
| 2 | 2 | **0.95** | **0.95** | **0.95** |
| 3 | 3 | 0.94 | 0.94 | 0.95 |
| 4 | 4 | 0.92 | 0.93 | 0.93 |
| 5 | 5 | 0.91 | 0.90 | 0.90 |
| 7 | 7 | 0.90 | 0.89 | 0.90 |

These insights further validate our Graph Sampling method as an effective approach for constructing graphs in a way that optimally captures the essential relationships within histological images, thereby improving the overall model performance.

### A.2.2 EFFECT OF MODEL ARCHITECTURE DEPTH

To explore the effect of different numbers of Graph Convolution (GC) blocks and Pseudo-Label Attention (PLA) blocks on the performance of a model designed for graph-based tasks, we conducted an experimental study. The results presented in Table 5 demonstrate a clear relationship between model depth and classification performance. The introduction of 2 GC blocks improves all metrics by approximately 0.01-0.03 points, suggesting that graph convolution operations effectively capture topological information from the input data.

Notably, the incorporation of PLA blocks leads to significant performance gains. The configuration with 2 GC blocks and 2 PLA blocks achieves optimal results across all metrics. This finding suggests that the combination of graph convolution and pseudo-label attention mechanisms creates a synergistic effect for tissue type classification.

However, deeper architectures with more than 3 blocks per type show diminishing returns, with performance gradually decreasing as model depth increases. The 7-block configuration even underperforms the baseline model, likely due to overfitting or optimization difficulties in very deep graph networks. This observation aligns with previous studies on graph neural networks (Wu et al., 2023; Xing et al., 2024), where excessive depth can lead to oversmoothing of node features. The results indicate that a balanced architecture with 2-3 blocks of each type provides the best trade-off between model capacity and generalization ability for this particular task.

### A.3 DETAILS OF DATASET PATH-DT-MSU WSS2V2

This study utilizes the PATH-DT-MSU WSS2v2 dataset, a publicly available collection of whole-slide histological images with polygonal annotations of tissue types. The dataset was compiled by expert teams from academic and medical research centers specializing in image processing and pathology[3].

Though the PATH-DT-MSU WSS2v2 dataset consists of only 10 whole-slide images, each was acquired at 40× optical magnification with resolutions exceeding 110,000 × 90,000 pixels, ensuring exceptionally fine-grained detail. The annotations focus exclusively on "clear" regions where pathologists can confidently delineate tissue boundaries, resulting in high-quality labels for four tissue types (AT, LP, MM, TUM) and background (BG). The areas of the annotated regions of the training sample are 1560, 7098, 533, 895, 1303 million pixels for classes AT, BG, LP, MM, TUM respectively at x40 magnification. The areas of the annotated regions of the test sample are 1086, 8032, 318, 743, 1199 million pixels for classes AT, BG, LP, MM, TUM respectively at x40 magnification. This selective annotation strategy prioritizes precision over quantity, compensating for the modest number of slides with unparalleled per-image information density. However, the dataset exhibits significant class imbalance, particularly in the training set where the background (BG) class

---

[3]Available at: [URL withheld for anonymity]

dominates with 7098 million pixels – approximately 13.3 times larger than the smallest class (LP at 533 million pixels).

### A.4 Training Details and Hyperparameters

#### A.4.1 Implementation Setup and Hardware

Our WSI-GT implementation utilized Python 3 with PyTorch and trained the model on a single NVIDIA A6000 48GB. For efficient graph processing, we constructed subgraphs each containing **64 patches** as the basic processing units following previous work (Li et al., 2023; Zheng et al., 2022). This patch count was empirically determined to balance computational efficiency and sufficient local context preservation.

#### A.4.2 Training Configuration

The training process was divided into two distinct phases, both conducted on the PATH-DT-MSU WSS2v2 dataset.

**First Phase:** This initial stage focused on training the patch-based encoder to enhance its ability to capture relevant features. We adopted a ResNet50 backbone initialized with ImageNet pre-trained weights to benefit from transfer learning and accelerate convergence. We employed the Adam optimizer with an initial learning rate of $1 \times 10^{-4}$. The learning rate followed a cosine warm-up strategy over 50 training epochs.

**Second Phase:** In this subsequent stage, we trained the WSI-GT model to effectively process the graph structure. This phase similarly used the Adam optimizer, but with a reduced initial learning rate of $1 \times 10^{-5}$. The training spanned 30 epochs with early stopping regularization to prevent overfitting, while maintaining the cosine warm-up learning rate schedule.

For training on Cell-Level and Universal Graph Node Classification Benchmarks, we follow all the instructions with original repositories (Vanea et al., 2022; Wu et al., 2023) for fair comparison.

#### A.4.3 Data Augmentation

We employed comprehensive data augmentation techniques to enhance model generalization when training on the histological PATH-DT-MSU WSS2v2 dataset. All augmentations were applied at the patch level and included:

- **Geometric transformations:** random horizontal and vertical flipping, and rotation (±15°)
- **Photometric adjustments:** contrast variation (scale factor 0.8–1.2)
- **Color jittering:** brightness adjustment (±20% of the original value)

#### A.4.4 Network Architecture Overview

The model architecture comprised the following key components:

**Patch-based Encoder** We adopted ResNet50 as our backbone network, initialized with ImageNet pre-trained weights. This choice provided a strong foundation for feature extraction while benefiting from transfer learning.

**Graph Transformer (WSI-GT)** WSI-GT integrates:

- **2 Graph Convolution (GC) blocks** for local neighborhood feature aggregation,
- **2 Pseudo-Label Attention (PLA) blocks** to capture intra-class dependencies while preserving spatial awareness.

This combination enabled efficient processing of graph-structured data while preserving important topological relationships in the histopathology images.

### A.5  SEMANTIC SEGMENTATION VISUALIZATIONS

This section presents representative visualizations of semantic segmentation results produced by our WSI-GT model on the PATH-DT-MSU WSS2v2 test set. Figures 7, 8, and 9 show comparisons between WSI-GT predictions, a fine-tuned ResNet50 baseline, and expert-annotated ground truth. These examples qualitatively illustrate that WSI-GT tends to produce more coherent and spatially consistent segmentations, particularly near tissue boundaries, compared to the baseline model.

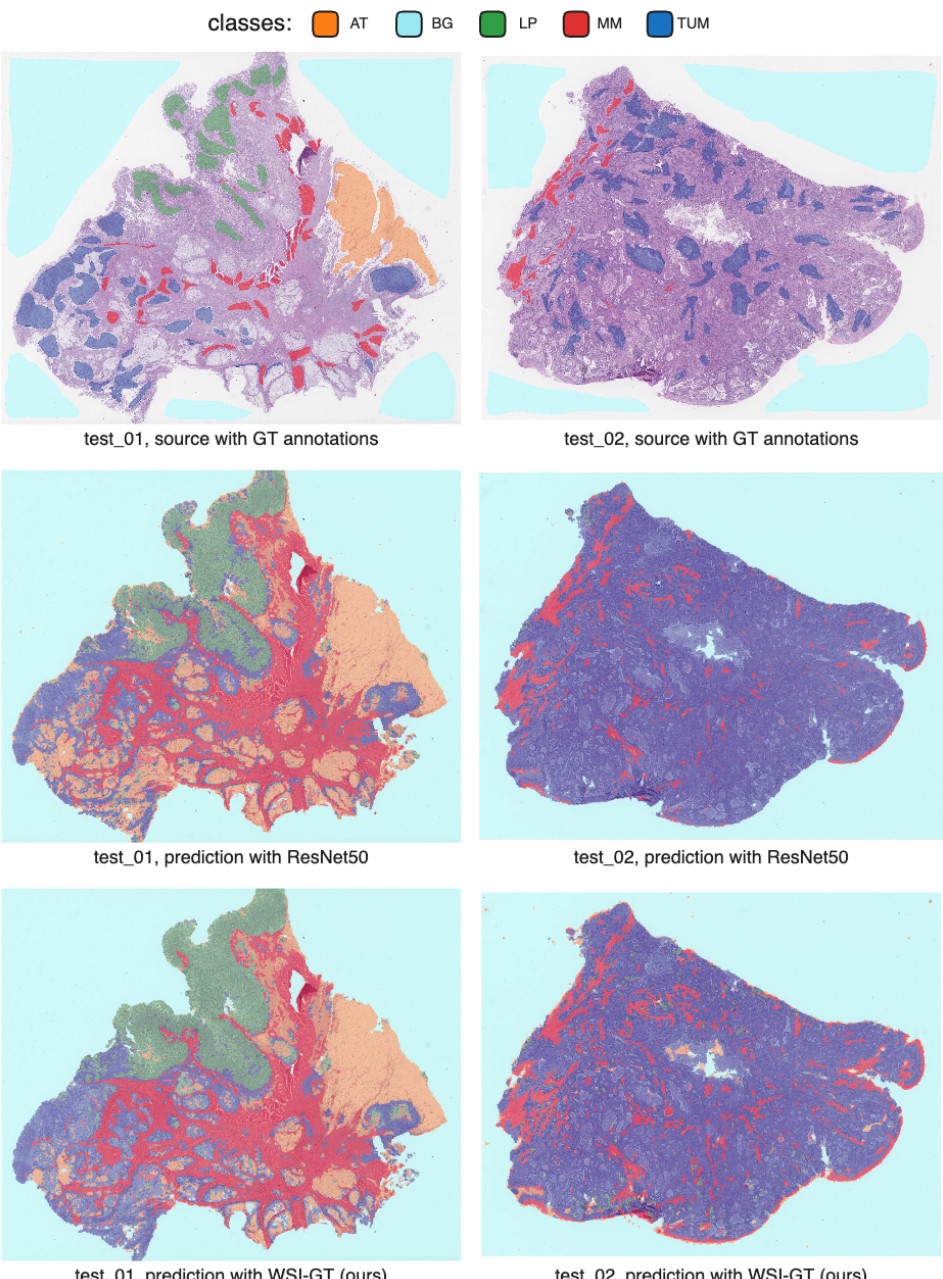

Figure 7: Comparative visualization of semantic segmentation results on the test set of PATH-DT-MSU WSS2v2 (test_01, test_02 images). The ground-truth polygonal annotations were made by expert pathologists.

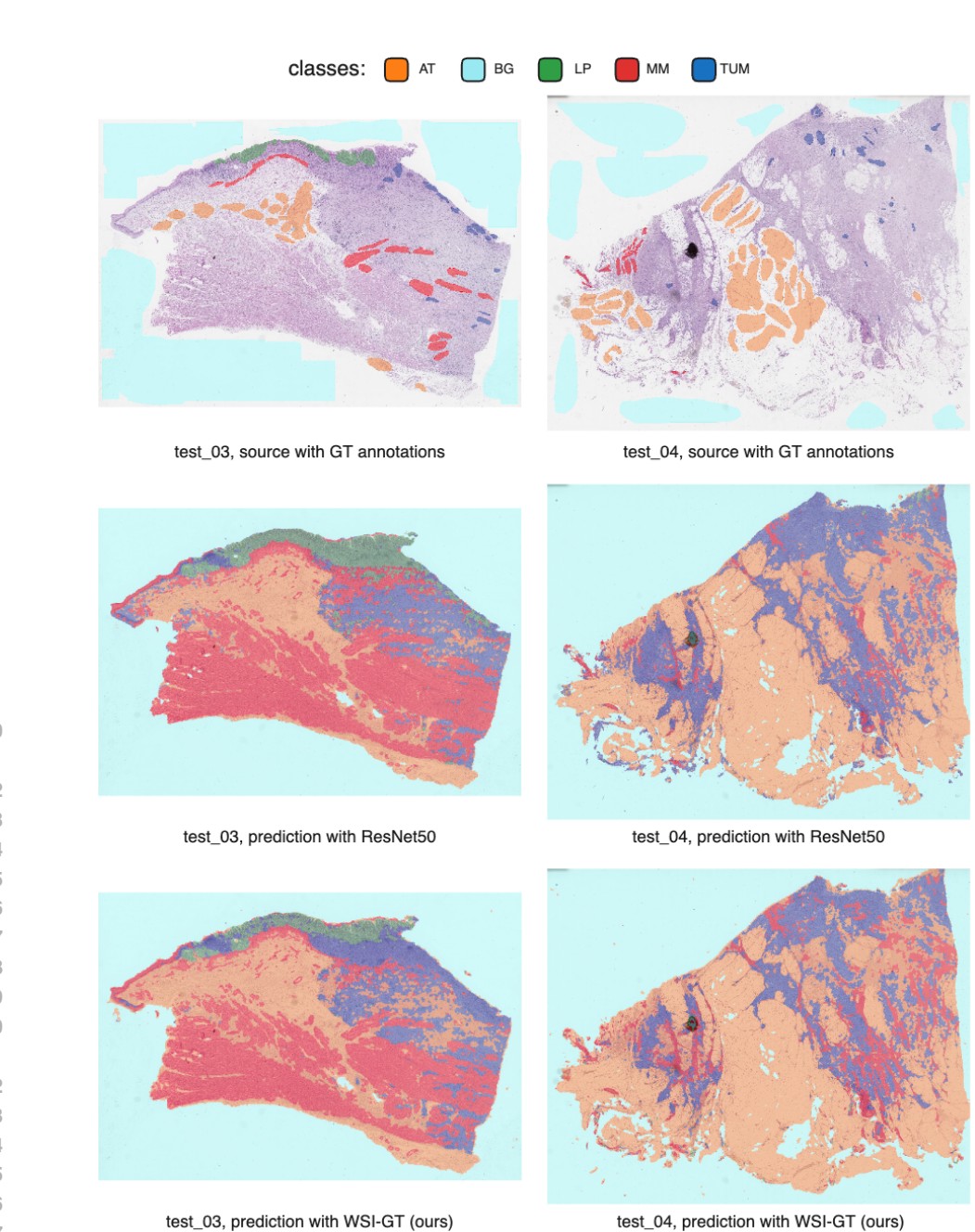

Figure 8: Comparative visualization of semantic segmentation results on the test set of PATH-DT-MSU WSS2v2 (test_03, test_04 images). The ground-truth polygonal annotations were made by expert pathologists.

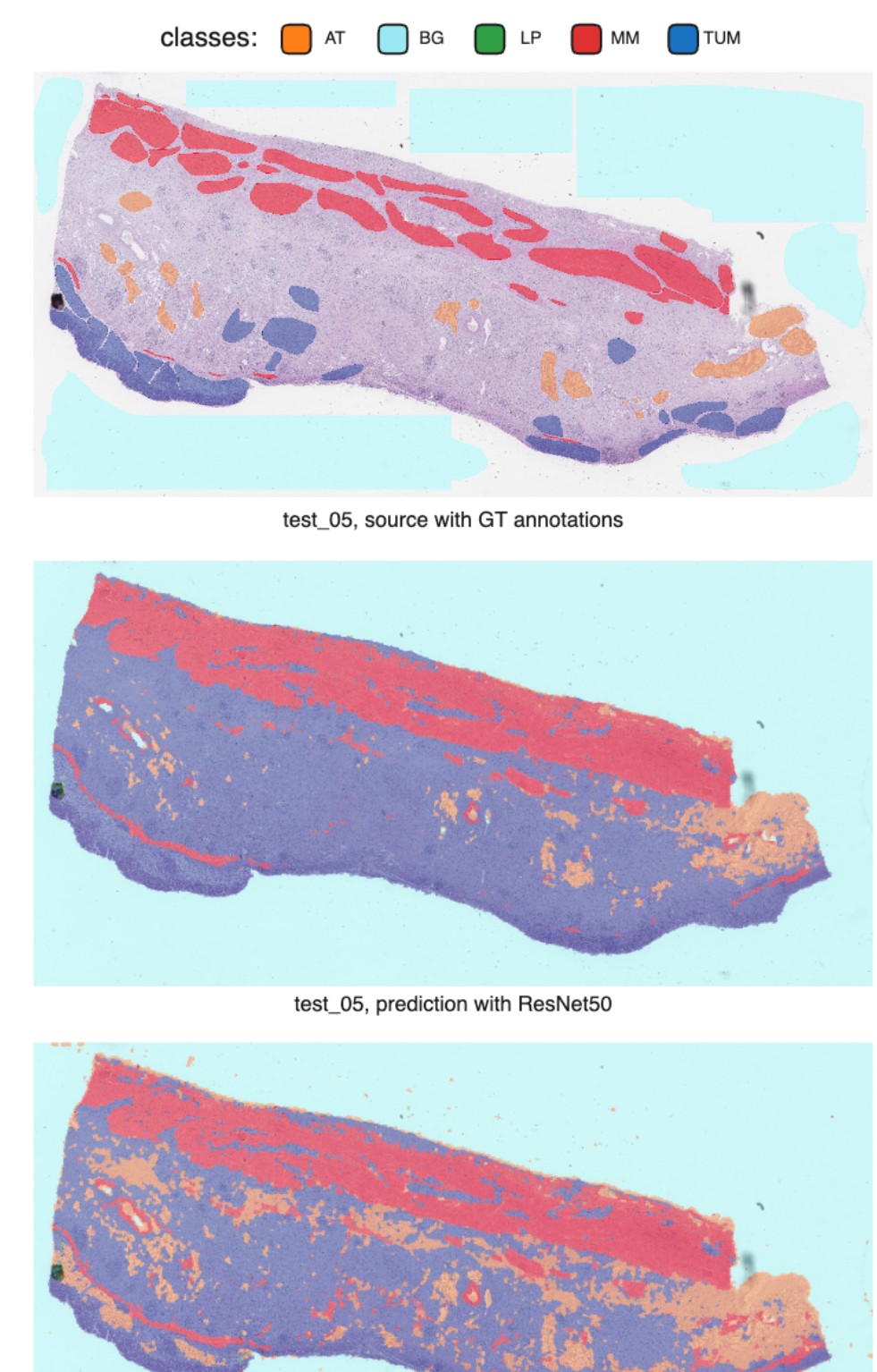

Figure 9: Comparative visualization of semantic segmentation results on the test set of PATH-DT-MSU WSS2v2 (test_05 image). The ground-truth polygonal annotations were made by expert pathologists.

