# OpenReview forum: "WSI-GT: Pseudo-Label Guided Graph Transformer for Whole-Slide Histology"
_ICLR.cc/2026/Conference — ICLR 2026 Conference Withdrawn Submission_

### Official Review · Reviewer_g2ci · 2025-10-30

**Soundness:** 2
**Presentation:** 3
**Contribution:** 2
**Rating:** 0
**Confidence:** 4

**Summary:**

This paper proposed a shallow graph to WSI patch classification. The topic is relevant and aligns with current efforts to leverage graph representations for digital pathology.

**Strengths:**

The paper addresses an important and timely problem in computational pathology with nice visualizations.

**Weaknesses:**

1.	Questionable state-of-the-art claims: Several existing shallow graph models, such as GPT [4], HEAT [5], and attention-based spatial models like CAMIL [3], are not adequately discussed or compared. While GTP [4] is mentioned in the introduction, it is not included in the experiments, which limits the validity of the SOTA claim.
2.	Incomplete baseline selection: strong CNN-based baselines such as DSMIL[1] and CLAM [2] are not mentioned at all. CLAM[2] is especially well established and can be easily extended to multi-class tasks. And the baselines mentioned in bullet point 1 are not included either.
3.	Questionable benchmark dataset: The proposed model is designed for WSIs, yet the datasets used—PubMed (Yang et al., 2016), Actor, and Deezer (Lim et al., 2021), are unrelated to histopathology or WSI classification. This discrepancy raises questions about the generalizability and practical relevance of the results.
4.	Unsupported claims of scalability: The claim that the proposed method provides a “practical and scalable solution” is not sufficiently substantiated by the experiments or analysis. See question 1.


[1] Li, Bin, Yin Li, and Kevin W. Eliceiri. "Dual-stream multiple instance learning network for whole slide image classification with self-supervised contrastive learning." Proceedings of the IEEE/CVF conference on computer vision and pattern recognition. 2021.
[2] Lu, Ming Y., et al. "Data-efficient and weakly supervised computational pathology on whole-slide images." Nature biomedical engineering 5.6 (2021): 555-570.
[3] Fourkioti, Olga, et al. "CAMIL: Context-aware multiple instance learning for cancer detection and subtyping in whole slide images." arXiv preprint arXiv:2305.05314 (2023).
[4] Zheng, Yi, et al. "A graph-transformer for whole slide image classification." IEEE transactions on medical imaging 41.11 (2022): 3003-3015.
[5] Chan, Tsai Hor, et al. "Histopathology whole slide image analysis with heterogeneous graph representation learning." Proceedings of the IEEE/CVF conference on computer vision and pattern recognition. 2023.

**Questions:**

1.	The annotation-based graph construction cannot be applied during inference, so a sampling process is needed. What is the overhead here? What practical advantage does this design offer?
2.	The term pseudo-label is used ambiguously. How are these labels generated, and how do they differ from the ground-truth annotations? If they are derived from annotations, what constitutes the pseudo-label during inference?

---

### Official Review · Reviewer_CXWv · 2025-10-30

**Soundness:** 3
**Presentation:** 3
**Contribution:** 1
**Rating:** 4
**Confidence:** 4

**Summary:**

The paper introduces WSI-GT, a simple Graph Transformer with Pseudo-Label Attention for precise tissue fragment classification in WSIs. It combines a ightweight graph convolution block for local feature aggregation with a pseudo-label attention block that restricts self-attention to nodes sharing the same pseudo-label. The authors also adopt an existing area-weighted random sampling approach for balanced subgraph construction and designed an adaptive coverage sampler for efficiently and unbiasedly constructing graphs from unannotated WSIs. Evaluating on the PATH-DT-MSU WSS2v2 dataset, the authors report a Macro-F1 of 0.95, outperforming both CNN and graph baselines. Additional tests on the Placenta cell-graph benchmark and standard node-classification datasets (PubMed, Actor, Deezer) suggest that the model generalizes to other domains.

**Strengths:**

The paper is clear and easy to follow. The label-aware self-attention is
an intuitive way to curb over-globalization in graph transformers while
keeping the benefits of attention; the GC+PLA mixing is straightforward
and easy to adopt.


• The paper addresses practical pitfalls in slide-level graph construction and
offers an area-weighted sampling scheme.
• The approach generalizes beyond WSIs, showing competitive performance
on a cell-graph benchmark and standard node-classification datasets.

**Weaknesses:**

• Novelty is modest for ICLR: the main contribution is a practical combination of known components (GC + hard pseudo-label mask; adopted area-weighted sampling) plus a simple inference sampler, without new
learning principles or strong theoretical/robustness evidence, despite solid
empirical results on a small dataset.
• PLA relies on pseudo-labels predicted by a pretrained patch encoder (ResNet-50),
but the paper does not fully specify the pseudo-labeling procedure (argmax vs. thresholded labels, calibration, whether pseudo-labels are fixed or updated during training, and how leakage is avoided). Given that PLA enforces a hard attention mask (Eq. 6), these details are important for reproducibility and for assessing robustness.

• For PubMed/Actor/Deezer, the paper says it “adopted the original eval-
uation pipelines,” but it is unclear what provides pseudo-labels to create the PLA mask on these datasets.

• The core experiment is performed on only one WSI dataset. The authors
should have shown results on at least another WSI dataset.

**Questions:**

How are pseudo-labels produced- by the fine-tuned ResNet50’s argmax?

Any confidence threshold or temperature scaling? Are pseudo-labels re-
computed during training (e.g., every epoch) or fixed after pretraining?

Have you considered the effect of noisy pseudo-labels?

 For PubMed/Actor/Deezer, what are the pseudo-labels?

---

### Official Review · Reviewer_d3Nc · 2025-10-30

**Soundness:** 2
**Presentation:** 2
**Contribution:** 2
**Rating:** 0
**Confidence:** 4

**Summary:**

The authors propose WSI-GT (Pseudo-Label Guided Graph Transformer), a graph-based framework for whole-slide image (WSI) analysis that mitigates oversmoothing and leverages limited annotations. It integrates local graph convolutions with pseudo-label guided attention to preserve tissue heterogeneity and introduces an area-weighted sampling strategy for class balance. WSI-GT demonstrates advantage on various tissue type classification benchmark and generalizes to other datasets, demonstrating strong scalability and clinical applicability.

**Strengths:**

The authors proposes a new graph neural network (GNN) architecture to address the tissue type classification, which is rarely addressed in the previous GNN applications on digital patholgoy

**Weaknesses:**

This work missing some key comparison as illustrated below

- It is clear that this work is just an extension from this published paper: "Tissue type classification for whole slide histological images with graph convolutional neural network", Sun et al. ,ICBSP 2024.  The only differences are pesudo-label attention block section in methodology and 2 additional evaluations in the experiments. However, there is zero discussion and comparison to the ICBSP 2024 work. If we directly quote the number from the ICBSP work on PATH-DT-MSU WSS2v2 data, The improvement from the additional  pesudo-label attention block is 0.03 (0.92->0.95). Also, it is weird that the number of other baselines (e.g., SGFormer) are different between ICBSP 2024 work (0.91) and this one (0.93). Ideally, they should be the same since the evaluation data is the same, which make all results suspicious.


- Another missing comparison is placenta benchmark by Vanea et al. 2022. The author completely missed their results in table 2, which i think is a very serious mistakes. Here are the quoted results from Table 1 in Vanea et al. 2022:
Classification accuracy: 67%
ROC AUC: 0.898
Both are "higher" than the results presented in this work (64.98 and 0.888), which undermine the sota claim from author.

**Questions:**

- Why did the author ignore the ICBSP 2024 work? Are author genuinely unaware of the existence of that?
- Why did the author not cite the results from Vanea et al. 2022 while saying: "adopted the original evaluation pipelines, integrating our method to ensure comparability with prior results?"

---

### Official Review · Reviewer_S1xr · 2025-11-04

**Soundness:** 2
**Presentation:** 2
**Contribution:** 1
**Rating:** 2
**Confidence:** 3

**Summary:**

This paper proposes a Graph Transformer approach for tissue fragment classification in pathology whole-slide images (WSIs). Its key idea is to construct a slide-level graph for classification, so as to properly process adjacent patches that may share the same label. To construct it, a sampling technique is proposed to obtain patches from multiple annotated regions. The experiments on PATH-DT-MSU WSS2v2 dataset show that the proposed graph transformer-based approach can surpass several compared baselines. On the other three datasets, the proposed approach often demonstrates better performance in classification tasks.

**Strengths:**

- The proposed approach is described clearly, with several illustrations and equations to show its key steps.
- Node-level classification tasks are also presented in the experiments, which could further demonstrate the generalization ability of the proposed approach.

**Weaknesses:**

I have several concerns as follows:
- The target task this paper focuses on is tissue fragment classification in pathology WSIs. However, only one dataset is used to evaluate the performance of the proposed approach. The authors are encouraged to validate their approach on more datasets in this field.
- Node-level classification experiments are conducted to confirm the generalization ability of the proposed approach. However, many state-of-the-art baselines are missing, e.g., HACT (Pati et al., Hierarchical graph representations in digital pathology, MedIA, 2022). The authors are encouraged to compare with recent state-of-the-art networks to show the strength of their methods.
- In Table 2, compared to GraphSAINT-rw, the proposed method fails to show advantages (only marginal improvements over the baseline) in node-level classification.

**Questions:**

- Could the authors use more datasets (instead of only PATH-DT-MSU WSS2V2) to benchmark the performance of all compared networks?
- Could the authors explain why the proposed method cannot perform in node-level classification?

---

### Note · Authors · 2025-11-24

I have read and agree with the venue's withdrawal policy on behalf of myself and my co-authors.